# A Review of Clinical Outcomes, Owner Understanding and Satisfaction following Medial Canthoplasty in Brachycephalic Dogs in a UK Referral Setting (2016–2021)

**DOI:** 10.3390/ani13122032

**Published:** 2023-06-19

**Authors:** Amy L. M. M. Andrews, Katie L. Youngman, Rowena M. A. Packer, Dan G. O’Neill, Christiane Kafarnik

**Affiliations:** 1Queen Mother Hospital for Animals, The Royal Veterinary College, Hawkshead Lane, Hatfield AL9 7TA, Hertfordshire, UK; kyoungman18@rvc.ac.uk (K.L.Y.); ckafarnik@rvc.ac.uk (C.K.); 2Department of Clinical Science and Services, The Royal Veterinary College, Hawkshead Lane, Hatfield AL9 7TA, Hertfordshire, UK; rpacker@rvc.ac.uk; 3Department of Pathobiology and Population Sciences, The Royal Veterinary College, Hawkshead Lane, Hatfield AL9 7TA, Hertfordshire, UK; doneill@rvc.ac.uk

**Keywords:** brachycephalic breeds, brachycephalic ocular syndrome, corneal ulceration, medial canthoplasty, Pug

## Abstract

**Simple Summary:**

Flat-faced (brachycephalic) dogs have increased in popularity despite growing awareness that extreme facial conformation with very bulging eyes compromises their health. Medial canthoplasty (MC) is a commonly performed surgical procedure in flat-faced dogs to improve their facial anatomy with the aim of decreasing eye irritation and discharge and reducing the likelihood of painful ulceration on the surface of the eye (cornea). This study analysed data from flat-faced dogs that were recommended and/or underwent MC. In addition, a questionnaire of owners’ perceptions, including eye problems before and after the surgery, and overall satisfaction were collected. Nearly three-quarters of the dogs recommended surgery had a history of, or actively had, corneal ulceration. Under half of the dogs recommended MC went on to have surgery and three quarters of these were Pugs. MC significantly reduced eye discharge, frequency of owners cleaning around their dog’s eyes, eye irritation and corneal ulceration with minimal complications and high owner satisfaction. These results demonstrate the value of MC surgery to improve the quality of life of flat-faced animals.

**Abstract:**

Brachycephalic breeds have increased in popularity despite growing awareness of their predisposition to a wide range of conformation-related diseases. The extreme facial conformation of many popular brachycephalic breeds compromises their ocular surface health, increasing the risk of painful corneal ulceration. Medial canthoplasty (MC) is a surgical procedure to address ocular abnormalities in brachycephalic dogs, which are collectively referred to as brachycephalic ocular syndrome (BOS). This study retrospectively reviewed the records of dogs recommended MC at a referral hospital between 2016 and 2021. A questionnaire was designed to identify owners’ perceptions pre- and post-operatively. From 271 brachycephalic dogs recommended MC, 43.5% (118/271) underwent surgery and 72.0% (85/118) were Pugs. The majority of dogs (73.7%, 87/118) that underwent surgery had current or historical corneal ulceration. Follow-up was available in 104 dogs, of which 5.7% (6/104) had corneal ulceration post-operatively. Sixty-four owners completed the questionnaire and reported post-operative corneal ulceration in 12.5% of dogs (8/64), reduced ocular discharge (70.8%, 34/48), reduced ocular irritation (67.7%, 21/31) and less periocular cleaning (52.5%, 32/61). Owners were satisfied with the clinical (85.9%, 55/64) and cosmetic (87.5%, 56/64) outcome. In conclusion, MC has high clinical relevance for the surgical management of BOS, restoring functional conformation and improving the quality of life of affected dogs.

## 1. Introduction

Brachycephalic ocular syndrome (BOS) is described as bilateral ocular changes associated with poor skull conformation, of which some or all abnormalities are associated with brachycephaly in dog breeds [1]. These abnormalities include a shallow orbit in conjunction with euryblepharon leading to ‘scleral show’, entropion, trichiasis, exotropia, lagophthalmos, reduced corneal sensation and compromised tear film [1]. Although ophthalmologically abnormal, scleral show is considered anatomically normal and even desirable in the brachycephalic dog breeding and showing community, and is even normalised by inclusion in the breed standard of the Japanese Chin [2,3].

BOS is associated with increased risk of painful corneal ulceration which potentially can contribute to the decision for euthanasia [4]. Brachycephalic dog breeds are reported to have twenty times increased risk in corneal ulceration compared to non-brachycephalic breeds and have increased risk of deep stromal involvement [2,5]. This may be associated with persistent conformation abnormalities, compromised tear film and reduction in corneal sensitivity [2,5,6,7]. Corneal grafting procedures are over-represented in brachycephalic breeds [8,9,10,11,12]. Although corneal repair surgery (i.e., corneal xenograft, conjunctival pedicle graft, corneoconjunctival transposition) may salvage the globe at risk, long-term post-operative vision in brachycephalic dogs can be poor due to progression of corneal pigmentation following a corneal surgery [1,13].

Medial canthoplasty (MC) is a surgical procedure undertaken to reduce the clinical signs of canine BOS and improve the quality of life of affected animals [1,14]. MC aims to correct a range of eyelid abnormalities, including medial entropion, caruncle trichiasis, macropalpebral fissure, as well as aiding functional blinking, reducing exposure keratitis and the likelihood of subclinical and clinical proptosis [14,15,16,17,18]. MC was originally described by Jenson (1979) and has since been modified by various authors [14,15,19,20,21]. MC has also been described in humans to address exposure keratopathy and has successfully improved clinical signs of lagophthalmos and ocular surface disease in 85% and 90% of human cases, respectively [22].

Eyelid surgery to correct ocular conformation abnormalities has been reported in only 8% of three common brachycephalic breeds (French Bulldog, Pug and English Bulldog), leading to the assumption that owners and primary-care veterinarians ‘normalise’ brachycephalic conformational abnormalities and consequently underuse surgical correction to improve the quality of life of these dogs [23].

The anatomical ocular abnormalities of certain brachycephalic breeds are raising increasing concern, particularly given the dramatically rising ownership of certain breeds over recent years [18,23]. According to UK Kennel Club statistics, French Bulldog registration increased by 3104%, Pug by 193% and English Bulldog by 96% between 2007 and 2017 [18]. The ocular abnormalities in these breeds have been identified as inherited and directly related to desired body shapes; therefore, breed standards should discourage exaggerated conformational abnormalities over time, striving to restore functional eye and eyelid anatomy and reduce the prevalence of clinical implications [24,25]. Previous empirical studies of dog conformation identified several key risk factors for poor corneal health, including the presence of nasal folds, short faces (muzzle less than half of the length of the cranium), wide eyelid apertures and exposed sclera (eye white) [2,26]. For some breeds, e.g., the Pug, some or all of these features are commonly present and result in a high risk of corneal ulceration. The risk of traumatic corneal ulceration secondary to facial pruritis remains unknown but may increase in dogs with uncontrolled atopic dermatitis, a disease to which Pugs are predisposed [27,28]. O’Neill et al. (2016) provided a framework of health priorities to improve the welfare of Pugs by addressing their incomplete blink, prominent eyes and poor eyelid conformation [29]. Pigmentary keratitis is a corneal disease reported in 39.1–87.3% of Pugs that can lead to blindness if left untreated [30,31]. MC has been reported to be superior to topical therapy alone to manage corneal pigmentation (i.e., pigmentary keratitis) related to BOS [21]. Yi et al. (2006) reported resolution of trichiasis and entropion-associated epiphora in all dogs which underwent MC and demonstrated an improvement in ocular surface irritation [15,16].

While the evidence for a negative impact from BOS on health and welfare is strong, the appealingly large eyes and short nose of brachycephalic breeds are still considered desirable traits for many owners [6]. There is ongoing investigation to try to understand the emotional reasoning behind purchasing a breed with known disease predispositions detrimental to their welfare, including cultural (e.g., fashion) drivers, as well as biological drivers (e.g., the ‘cute effect’ due to their baby-like facial appearance) [32,33,34,35]. Previous studies have reported that 93% of current owners of brachycephalic dogs were likely to want to own the breed type again [36]. Therefore, it is likely that the popularity of brachycephalic breeds will persist internationally in the absence of relevant laws to restrict breeding and/or ownership. Therefore, first-opinion veterinary education to owners is essential for raising awareness of common health conditions in brachycephalic breeds, and the available treatments to improve the quality of life of affected dogs. As such, veterinary surgeons making owners aware that BOS can be improved by MC surgery may be key to maintaining and improving the quality of life of the current generation of brachycephalic dogs. This is with the caveat that a better long-term messaging from a canine welfare perspective is for owners to ‘stop and think before buying a flat-faced dog’ [37], to reduce their popularity and strive to exclude dogs with extreme skull conformations from breeding [36].

Although MC is a commonly performed surgery, post-operative outcome reports in the veterinary literature are sparse, with only one report to the authors’ knowledge to date [15].

The aims of this study were to identify the surgical outcomes and complications and owners’ perceptions of MC in a UK referral setting, alongside comparing the incidence of corneal ulceration before and after the eyelid surgery. In addition, the study aimed to explore owner awareness of ocular abnormalities in brachycephalic dogs before and after MC and determine owner satisfaction following this surgery.

It was hypothesised that the proportion of dogs with corneal ulceration, ocular discharge and ocular irritation based on owners’ reports are reduced after MC.

## 2. Materials and Methods

### 2.1. Retrospective Data

The Royal Veterinary College (RVC) VetCompass Programme was used to examine electronic patient information regarding dog breeds with brachycephaly that were recommended MC eyelid corrective surgery during a five-year period (2016–2021) [38]. Dogs under the care of the Queen Mother Hospital for Animals were defined as those with one or more electronic patient records during 2016 to 2021. VetCompass collates de-identified electronic patient record data from primary-care veterinary practices in the UK for epidemiological research. Data fields available to VetCompass researchers include a unique animal identifier along with species, breed, date of birth, sex, neuter status and also clinical information from free-form text clinical notes, summary diagnosis terms and treatment with relevant dates [39]. The MC case definition required evidence in the clinical records that the dog was recommended MC in one or both eyes at any date from 1 January 2016 to 31 December 2021. The study included ten breeds (Table 1) that were described as brachycephalic according to the Kennel Club and/or are reported to have inherited eyelid conditions according to the European College of Veterinary Ophthalmologists [24,40].

A total of 494 candidate animals were manually reviewed. All clinical records were searched for potential inclusion using the clinical free text field (‘*medial canthoplasty*’, ‘*medial canth**’, ‘*medial canthoplasties*’) [41]. The breeds of interest were selected during the time period of interest. The clinical records of dogs with brachycephaly were then manually reviewed following recommendation of MC.

Breed, sex and neuter status were defined according to the status in the final available clinical record. Age (years) was defined based on the first documented date an MC was recommended. Age was further categorised as: ≤1.0, 1.0 to <2.0, 2.0 to <4.0, 4.0 to <6.0, 6.0 to <8.0, 8.0 to <10.0 and ≥10.0 years [39].

Cases recommended for MC were categorised into those that did (‘surgical MC group’) and did not (‘non-surgical MC group’) undergo MC. MC was defined as eyelid shortening with or without modification (i.e., additional triangular modified Celsus–Hotz surgery), resulting in a reduction of the palpebral fissure at the medial canthus and corrected eyelid position (entropion) [42]. An excessive palpebral fissure is classified as eyelid length greater than 6–8 mm than the corneal diameter [43]. All surgery was performed at the Queen Mother Hospital for Animals (QMHA) by an ECVO board-certified ophthalmologist and/or resident-in-training. For each case, the following information was collected: signalment, history of corneal ulceration, ophthalmic abnormalities for surgical MC and non-surgical MC group. Information from the surgical MC group was collected regarding if surgery was planned or emergency surgery with concurrent corneal repair surgery, the period between recommendation and surgery, concurrent ocular or other surgery, post-operative ophthalmic abnormalities, post-operative (short- and long-term) medical management, time from surgery to follow-up examination in days, complications and occurrence of post-operative corneal ulceration.

Information on ocular abnormalities were extracted from the clinical histories noted in any ophthalmic examination by the ophthalmologist (board-certified or resident-in-training). Dogs with pre-existing diagnosis of atopic dermatitis were identified.

A complication of surgery was associated with any negative change to the immediate post-operative appearance of the eyelids which raised concern to the owner or ophthalmologist (i.e., wound dehiscence, suture loss, wound infection, wound breakdown, suture ends causing corneal ulceration). All owners consented to the use of their data by completion of the consent form on admission to the hospital.

### 2.2. Owner Questionnaire of Surgical MC Group

An online owner questionnaire was designed to identify perceptions of ocular abnormalities before and after surgery, long-term medications, post-operative complications and satisfaction following surgery. Ethical approval was obtained from the Royal Veterinary College Social Sciences Research Ethical Review Board (URN: SR2022-0005).

The original referring primary-care practice was contacted prior to contacting the owner to ensure the patient was still alive. Owners were contacted via telephone and were informed of the aims of the study and were emailed an online survey link hosted by SurveyMonkey^®^ (valid for two months) or received a postal copy. A paper-based option for the questionnaire was also offered to increase accessibility.

#### Questionnaire Design

Section 1: *Owner perspective of preoperative ocular abnormalities*. This included four closed questions (yes/no/unsure) exploring the presence of ocular discharge with further multiple-choice questions regarding its characteristics (colour and consistency), their ocular hygiene regimen, signs of ocular irritation (i.e., rubbing, pawing or itching eyes/eyelids) and previous history of corneal ulceration.

Section 2: *Owner awareness of brachycephalic conformation and ocular abnormalities in general and in their own dog*. Owners reported (yes/no) if they were aware that their dog’s face/head/eye shape was associated with increased risk of eye disease and if they were aware their own dog had eyelid abnormalities [1].

Section 3: *Reason for specialist referral and response to MC recommendation*. Owners were asked six varying open and closed questions regarding the reasons for referral to an ophthalmic specialist, the recommendation of MC, the duration before surgery was pursued, their reservations for surgery, concurrent surgery performed at the time of MC and their previous awareness of this type of eyelid surgery.

Section 4: *Surgical complications*. Owners were asked to state if there were any post-operative complications (yes/no) and to describe them using free text. Owners reported who primarily managed these complications (the study centre (QMHA)/their primary-care veterinarian/themselves with at-home care).

Section 5: *Owner perspective of postoperative ocular abnormalities*. This repeated the initial four closed questions (yes/no/unsure) to identify the presence of ocular discharge with further multiple-choice questions regarding its characteristics (see Section 1), their ocular hygiene regimen and any corneal ulceration since MC.

Section 6: *Long-term ocular medical management*. Owners reported if long-term ocular medication was used (yes/no) and if so, what medication had been continued.

Section 7: *Owner satisfaction following MC*. Owners reported whether they were satisfied with the clinical and cosmetic outcome following surgery (yes/no/unsure). If they were not satisfied, they were given the opportunity to explain the reason(s) in free text. Furthermore, owners were asked (yes/no/unsure) if they would consider another brachycephalic dog undergoing this procedure in the future (if they were to own one and it was recommended), whether they would recommend this surgery to other owners of dogs where it was indicated and if not, to explain the reason(s).

### 2.3. Statistical Analyses

For statistical analysis, all data were exported from VetCompass and Survey Monkey^®^ (Momentive Inc., San Mateo, CA, USA) into an Excel spreadsheet (Microsoft Corporation, Redmond, WA, USA, 2022) and manually reviewed [38]. Data were then imported to IBM SPSS Statistics version 28 (IBM Corp., Armonk, NY, USA) to perform basic descriptive statistics.

Mean values were used for normally distributed data; otherwise, median values were reported. Categorical data were analysed using a two-tailed chi-squared with Yates correlation test to test the hypothesis. Risk factor analysis included all cases that underwent surgical MC compared to non-surgical MC dogs [41]. Binary logistic regression modelling evaluated univariable associations between risk factors (age, sex, neuter status, breed) and the outcome of being a dog undergoing MC between 2016 and 2021. The youngest age group was used as the reference. The breed used as reference was Boston Terrier and was chosen due to it being the median group of representation. Statistical significance was considered as a *p* value < 0.05. Breeds with low sample size (*n* = 1) (Chihuahua, Japanese Chin, Lhasa Apso, Pekingese) could not be included in models exploring the impact of breed.

## 3. Results

### 3.1. Retrospective Analysis

A total of 29.1% of dogs (271/931) spread across nine breeds with brachycephaly were recommended an MC from 1 January 2016 to 31 December 2021 (Table 1). The Pug was the most represented breed (71.6%, 194/271), followed by the Shih Tzu (17.7%, 48/271), French Bulldog (5.5%, 15/271), Cavalier King Charles Spaniel (2.6%, 7/271) and Boston Terrier (1.1%, 3/271) (Figure 1, Table 2). There was one dog representing each of the following breeds: Chihuahua, Lhasa Apso, Japanese Chin and Pekingese. All other breeds with brachycephaly investigated in this study were recommended MC in less than a third of the breed population seen during the same five-year period. Chihuahuas were the breed least likely to be recommended an MC (Table 2). The mean age for recommendation of MC was 4.5 years (SD ± 3.04 years). The age category of 2.0 to less than 4.0 years had the most patients (23.6%, 64/271) (Table 3). The recommended cases included 42.4% females (115/271), of which 65.2% were neutered (75/115), and 57.6% males (156/271), of which 53.2% were neutered (83/156).

Previous or current corneal ulceration had been diagnosed in 76.0% of dogs (206/271), with 44.2% (91/206) of these dogs having experienced two or more episodes of corneal ulceration. In contrast, 21.0% (57/271) had no concurrent corneal ulceration and 3.0% (8/271) were without history of corneal ulceration according to the medical history.

MC was recommended in a routine ophthalmology referral examination in 88.9% of dogs (241/271); the remainder were associated with emergency referral for corneal ulceration (11.1%, 30/271). The three most common additional ocular abnormalities associated with BOS were recorded as entropion with medial lower eyelid trichiasis (85.1%, 205/241), euryblepharon (72.6%, 175/241) and pigmentary keratitis (68.9%, 166/241) (Table 4). Fifteen percent of dogs (15.5%, 42/271) had a history of atopic dermatitis, of which over fifty percent of which (54.8%, 23/42) were receiving topical and/or systemic immunomodulatory medication at the time of presentation.

#### 3.1.1. Non-Surgical MC Group

Although MC had been recommended, there was no evidence in the clinical records that the surgery was performed in 56.5% of dogs (153/271). The reasoning of owners in not pursuing MC was not recorded in the majority of reports but age of the patient (6.5%, 10/153), comorbidities (3.3%, 5/153), concerns about anaesthetic risk (2.6%, 4/153) and financial constraints (2.6%, 4/153) were cited in others. The owners of six dogs (2.2%, 6/271) initially requested but did not pursue surgery.

Pugs were the most common breed (71.2%, 109/153), followed by Shih Tzus (9.8%, 15/153) and French Bulldogs (7.8%, 12/153). Under fifty percent of these Pugs (47.7%, 52/109) and forty-eight percent of non-Pugs (48.5%, 16/33) had corneal ulceration at the time of MC recommendation (Table 4). Concurrent ocular diseases in Pugs were corneal pigmentation (82.6%, 90/109), entropion (93.4%, 102/109) and euryblepharon with increased scleral show (72.5%, 79/109). Non-Pug breeds had euryblepharon with increased scleral show (72.7%, 32/44).

In total, 12.4% of dogs (19/153) re-presented to the referral hospital for further episodes of corneal ulceration and 87.6% (134/153) of dogs were lost to follow-up. Under half of these dogs then underwent MC (42.1%, 8/19) at the referral hospital after a median time of 150 days (IQR 104–329, range: 34–718) following recommendation and were included in the surgical MC group. Two percent of dogs (2.0%, 3/153) presented to the referral hospital for two or more further episodes of corneal ulceration but MC was not pursued.

#### 3.1.2. Surgical MC Group

MC was performed in 43.5% (118/271) of recommended patients. Of these, surgery was most often bilateral (83.9%, 99/118), and was unilateral in 9.3% of cases (11/118). Of the dogs that had unilateral surgery, 2.5% of dogs (3/118) underwent enucleation of the contralateral eye and 2.5% of dogs (3/118) underwent corneal reconstructive surgery. Four percent of dogs (4.2%, 5/118) had already undergone unilateral enucleation prior to presentation. Additional surgical procedures associated with BOS performed at the same time as MC included nasal fold resection (11.9%, 14/118), corneal grafting (11.0%, 13/118), additional modified Celsus–Hotz in the ventromedial eyelid (10.2%, 12/118), distichia (5.1%, 6/118) and ectopic cilia removal (0.8%, 1/118). Concurrent brachycephalic obstructive airways syndrome surgery was performed in 19.5% of dogs (23/118). The median time from MC recommendation to surgery was 44 days (IQR 14–100, range: 0–1314).

It was reported that 73.7% of dogs (87/118) that underwent MC had current or previous corneal ulceration, of which a current corneal ulcer was present in 18.6% of dogs (22/118) (Figure 2C) and a healed corneal ulcer in 55.1% of dogs (65/118) (Figure 2D). Pugs were the most common breed in the MC surgical group (72.0%, 85/118), followed by Shih Tzus (19.5%, 23/118) and French Bulldogs (2.5%, 3/118). The following ophthalmic abnormalities were recorded for Pugs with concurrent corneal pigmentation (64.7%, 55/85), entropion (67.1%, 57/85) and euryblepharon with increased scleral show (50.6%, 43/85) (Table 4, Figure 2A). The non-Pug dogs had entropion (54.5%, 18/33) and, less often, corneal pigmentation (27.3%, 9/33). The percentage of Pugs with corneal pigmentation was statistically significantly higher than for non-Pug dogs (*p* < 0.001). There was no significant difference in the presence of entropion between Pugs and non-Pug dogs (*p* = 0.111).

#### 3.1.3. Post-Operative Analysis

The median duration of hospitalisation was one day (IQR 1–1, range 0–5). Post-operative topical medications included broad-spectrum antibiotics (77.1%, 91/118), 0.2% cyclosporine A (61.0%, 72/118), lubricants (43.2%, 51/118), 0.02% tacrolimus (9.3%, 11/118), serum (3.4%, 4/118) and non-steroidal anti-inflammatory drugs (NSAID) (2.5%, 3/118). Systemic non-steroidal anti-inflammatories were prescribed for 88.1% (104/118) and broad-spectrum systemic antibiotics were prescribed for 40.7% (48/118) of dogs. Long-term topical medication was recommended following MC including 0.2% cyclosporine A (51.7%, 61/118), lubrication (20.3%, 24/118), 0.02% tacrolimus (9.3%, 11/118) and topical NSAID (4.2%, 5/118) (Figure 3). These were prescribed according to the ophthalmologist’s preference and/or the severity of tear film deficiency and corneal pigmentation.

Follow-up information was available in 88.1% of dogs (104/118) with median re-examination at the referral hospital within 14 days (IQR 10–16, range: 4–179). Post-operative complications were reported in 10.6% of dogs (11/104) associated with suture breakdown or loss (5.8%, 6/104), suture reaction (3.8%, 4/104) and corneal ulceration from sutures (2.9%, 3/104). One dog (1.0%, 1/104) had suture breakdown and corneal ulceration and one dog (1.0%, 1/104) had suture breakdown and reaction. Two patients (1.9%, 2/104) underwent a second entropion correction surgery. Satisfactory eyelid position was reported in 69.2% of dogs (72/104) at the first post-operative examination; however, this was not commented on in the remainder of dogs.

A single episode of corneal ulceration not associated with the surgical sutures was reported following MC in five dogs (4.8%, 5/104), and one dog experienced two episodes (1.0%, 1/104). This was significantly lower than pre-operatively (*p* < 0.001).

Proportional corneal ulceration did not differ statistically between the non-surgical MC group (12.4%, 19/153) and the patients who underwent an MC (5.8%, 6/104) (*p* = 0.310).

### 3.2. Owner Questionnaire

Of the 118 dogs that underwent MC (2016–2021), 84.7% (100/118) that were confirmed to be alive and had been seen by the primary-care practice or QMHA since 1 June 2021 were invited to join the survey. Of these dogs, 64.0% of owners (64/100) completed the questionnaire (62 online questionnaires and 2 paper questionnaires). Owners re-locating (6.0%, 6/100) or having a change in home circumstances (2.0%, 2/100) limited completion of the questionnaire. A further two owners (2.0%, 2/100) started the questionnaire but did not complete it. Results are summarised in Table 5.

#### 3.2.1. Ocular Discharge and Periocular Cleaning

Pre-operative ocular discharge was reported by the majority of owners (75.0%, 48/64) and was significantly improved (70.8%, 34/48) post-operatively (*p* < 0.001), supporting the hypothesis. Ocular discharge was most often noted by the owners to occur on a daily basis pre-operatively (62.5%, 30/48) (Table 5).

Pre-operatively, the colour of the ocular discharge was reported to be clear (33.3%, 16/48), cream (20.8%, 10/48), white (20.8%, 10/48), brown (8.3%, 4/48), yellow (6.3%, 3/48,), green (6.3%, 3/48) or grey (2.1%, 1/48).

The texture of the discharge pre-operatively was reported to be mainly mucus-like (45.8%, 22/48). The texture of the ocular discharge was reported to be mucus-like (47.9%, 23/48), wet and sticky (22.9%, 11/48), watery (22.9%, 11/48), dry and crusty (4.2%, 2/48) or pus-like (2.1%, 1/48).

Periocular cleaning was reported by the majority of owners (79.7%, 61/64) and was most often required daily (42.6%, 26/61) (Table 5). Tap water was mainly used for cleaning (62.7%, 32/51).

Post-operative ocular discharge was reported in 65.6% of dogs (42/64), which was statistically significantly improved compared to before surgery (*p* < 0.001) (Table 5). Some owners reported the presence of ocular discharge remained the same (29.2%, 14/48) or increased (16.7%, 8/48). The characteristics of the discharge remained the same in 95.8% of dogs (23/24).

Periocular cleaning was performed on 83.6% of dogs (51/64) post-operatively. The requirement for periocular cleaning was significantly reduced compared to before surgery (*p* < 0.001, Table 5). Water remained the most common way of cleaning the periocular area (35.3%, 18/51).

#### 3.2.2. Ocular/Periocular Irritation

Owners reported signs of ocular/periocular irritation (itching, rubbing or pawing) pre-operatively in 48.4% of dogs (31/64). Ocular/periocular irritation was often reported daily (71.0%, 22/31) (Table 5), of which 21.9% of dogs (14/64) were diagnosed with concurrent atopic dermatitis, with 14.3% (2/14) receiving immunomodulatory medication prior to surgery.

Post-operatively, owners reported signs of ocular/periocular irritation (itching, rubbing or pawing) in 40.6% of dogs (26/64). Owners reported more than half of the dogs showed signs of ocular/periocular irritation less often than before surgery (67.7%, 21/31) (Table 5), of which 14.3% of dogs (3/21) had atopic dermatitis and were not receiving additional treatment. The level of ocular irritation was significantly lower than before surgery (*p* = 0.030), which supported the hypothesis.

#### 3.2.3. Corneal Ulceration

At least one episode of corneal ulceration had been reported pre-operatively in 68.8% of dogs (44/64) (Table 5). Post-operatively, 12.5% of dogs (8/64) had experienced at least one episode of corneal ulceration. Thirty-seven percent of these dogs had not had corneal ulceration pre-operatively (37.5%, 3/8).

The owners of 4.5% of dogs (2/44) reported that the occurrence of corneal ulceration in their dog was less than before surgery.

Of the 44 dogs that had corneal ulceration prior to surgery, 81.9% of dogs (36/44) had not experienced corneal ulceration following MC. The proportions of dogs with corneal ulceration pre-operatively were significantly higher than post-operatively (68.8% vs. 12.5%, respectively, *p* < 0.001), supporting the hypothesis of the current study.

#### 3.2.4. Owner Awareness of BOS and Response to MC Recommendation

Around half of owners (51.6%, 33/64) expressed awareness that their dog’s breed predisposed their dog to ocular disease, but only one fifth (20.3%, 13/64) expressed awareness that their own dog exhibited ocular abnormalities (Table 6).

MC was recommended on the initial consultation at the referral hospital in 53.1% of dogs (34/64), and following recommendation, 57.8% of owners (37/64) scheduled the surgery immediately. The remaining owners scheduled the surgery within a month (15.6%, 10/64), within six months (10.9%, 7/64) and one year (3.1%, 2/64). All other owners (12.5%, 8/64) could not recall the time period between recommendation and scheduling the surgery.

Over a third of owners (39.1%, 25/64) expressed reservations regarding pursuing surgery, including concerns about the anaesthetic (52.0%, 13/25), further damage to their dog’s eyes (24.0%, 6/25) and changes to their dog’s physical appearance (16.0%, 4/25).

Thirty dogs that underwent MC had another surgical procedure performed during the same anaesthetic (46.9%, 30/64). These included brachycephalic obstruction airway syndrome (BOAS) correction (26.7%, 8/30), enucleation of the contralateral eye (13.3%, 4/30), nasal fold resection (10.0%, 3/30) or other surgery (43.3%, 13/30). Only one quarter of owners (23.4%, 15/64) expressed awareness of the existence of MC as a surgical option before presentation to the referral hospital, and of these, 60.0% (9/15) had considered it for their own dog.

#### 3.2.5. Post-Operative Complications, Long-Term Treatment and Owner Satisfaction

Post-operative complications associated with MC surgery were reported by 15.6% of owners (15.6%, 10/64), most commonly reported as associated with the surgical site (30.0%, 3/10), gastrointestinal signs (30.0%, 3/10), BOAS-related complications (10.0%, 1/10), pain (10.0%, 1/10) and corneal ulceration (10.0%, 1/10). These complications were generally managed by the referring primary-care practice (50.0%, 5/10).

Over half of the dogs that underwent MC remained on long-term topical medication (54.7%, 35/64), most commonly 0.2% cyclosporine A (77.1%, 27/35), lubrication (14.3%, 5/35) and 0.02% tacrolimus (11.4%, 4/35) (Figure 3).

Overall, most owners were satisfied with the clinical (85.9%, 55/64) and cosmetic (87.5%, 56/64) outcome of MC. In addition, most owners would consider this surgery under veterinary recommendation for another dog in the future (85.9%, 55/64), with a slightly reduced number of owners (78.1%, 50/64) recommending the surgery to another owner in the same situation (Table 7).

## 4. Discussion

This is the first study of MC surgery in dogs to investigate clinical outcomes and owner perspectives of this surgery. The results demonstrated a significant improvement of most of the ocular abnormalities in a large proportion of patients that underwent surgery. These results support the hypothesis that the proportion of dogs with corneal ulceration, ocular discharge and ocular irritation (based on owners’ perceptions) are significantly lower after MC and that owners are satisfied with the surgical outcomes.

The majority of dogs that were recommended MC were young-to-middle-aged dogs (2–6 years), with a trend towards male dogs being more likely to be recommended MC than female dogs. Male dogs with brachycephaly have previously been reported to be significantly over-represented in other studies, and further investigation into this is warranted [36,44,45].

The study population showed an over-representation of Pugs, despite the French Bulldog being the most commonly registered breed with the Kennel Club each year during the study period, from 2016 to 2021 [26,46]. This over-representation supports the study by O’Neill et al. (2016), which reported ophthalmological disorders to be the most prevalent in 16.25% of Pugs within general practice [29]. Pugs have an ultra-predisposition to corneal ulceration, as reported by O’Neill et al. (2022), with thirteen times greater odds of developing corneal ulceration compared to non-Pugs [47]. This may be associated with their extreme exaggerated reduction in craniofacial ratio (reflected in the degree of facial ‘flattening’), which has resulted in an increased risk of corneal ulceration [2].

Although pigmentary keratitis was not a focus of the current study, the high proportion of Pugs may also be attributable to their predisposition for pigmentary keratitis and the fact that MC aids to prevent progression [15,30]. These predispositions to ocular disease, alongside the dramatic increase in Pug ownership over the last decade, demonstrate the importance of awareness of the value of MC within multi-modal clinical management to reduce negative effects on the ocular health of these dogs [29]. Moving our shared human preference towards a desire for less extreme skull conformation in brachycephalic dogs may reduce the levels of BOS in Pugs and other brachycephalic breeds and reduce the requirement for remedial surgeries such as MC to redress ocular abnormalities [48]. Noted by the authors’ clinical observations, English Bulldogs were not recommended MC in this study, as they anatomically require an alternative type of blepharoplasty to correct entropion, ectropion and/or macroblepharon. Breed representation in the referral setting is skewed to that in primary-care veterinary practice, as often, normalisation of clinical signs associated with extreme brachycephaly occurs [49].

The percentage of patients (85.1%, 205/241) recommended MC for entropion was similar in the present study to that reported by Yi et al. (2006) [15]. Their study population differed, with Shih Tzus being the most prevalent breed (65.2%, 15/23), and with MC being primarily performed to manage epiphora associated with medial canthal trichiasis (91.3%, 21/23) and/or medial canthal entropion (82.6%, 19/23). In the same study, the history of corneal ulceration (4.3%, 4/23) and pigmentary keratitis (39.1%, 9/23) was less compared to the current study (76.0% and 68.9%, respectively). This may be associated with the variation in sample size (23 dogs vs. 118 dogs), breed representation, severity of conformation abnormalities or differing stage of disease between the two study populations. The increase in ownership of dogs with extreme brachycephaly over the recent decade may also be a contributing factor [18].

It is noteworthy that over half the owners recommended MC did not pursue surgery in the current study. The reasons remained largely unknown; however, the age of the dog and general anaesthetic risk were recorded factors in the clinical notes. Concern for anaesthetic risk was also reported by owners with dogs that underwent MC. These reservations may have also increased the time period between MC recommendation and surgery. Several studies have documented higher anaesthetic risk in brachycephalic dogs compared to non-brachycephalic dogs [50,51]. In the present study, only one patient (1.6%, 1/64) was reported by the owner to have experienced a post-operative complication associated with their breathing, which is lower than what has been reported in other literature [50,52]. This may also reflect the specialist anaesthesia service in a referral setting. Increased anaesthetic risk has been demonstrated in brachycephalic dogs with longer anaesthetic duration [50]. Therefore, MC should be considered as a sole procedure to minimise anaesthetic time and not combined with BOAS or other surgeries where possible.

Although more than half of the owners were aware that their dog’s breed predisposed their dog to ocular disease, only 20.3% of owners (13/64) acknowledged that their own dog exhibited ocular abnormalities. This is similar to a previous study of brachycephalic dogs in the same referral hospital, in which 58% of owners who reported that their dog exhibited clinical signs of BOAS did not consider their dog to have a ‘breathing problem’ [49]. It is likely that these disconnects reflect a degree of ‘cognitive dissonance’—in this case, psychological discomfort caused by the knowledge that their dog’s breed predisposes them to ocular disease, whilst also being the owner of a dog of that breed (presumably through choice). Packer et al. (2019) reported 63.1% of owners claiming that their brachycephalic dog was ‘healthier’ than the average dog of their breed, and that only 6.8% of owners considered their dog as either ‘less’ (5.3%) or ‘much less’ (1.5%) healthy than average for their breed, despite coming from a population of relatively young dogs with a high disease burden (including 15.4% having been diagnosed with a corneal ulcer, and 8% having undergone corneal surgery) [23].

The complication rate following MC noted in the clinical records (10.6%, 11/104) and by the owners (15.6%, 10/64) was higher compared to other elective veterinary surgeries (6.1%, 62/1016), including routine neutering and declawing [53]. Comparative complication rates of MC studies are lacking. However, from 104 patients with the eyelid position and entropion corrected, only two patients that underwent MC required further eyelid surgery, which is seen as an encouraging success of the surgery. Determining complications directly associated with MC was not possible due to the retrospective nature of the study.

The current study reports that a high proportion of owners continued to use the topical medications recommended for their dog, which was interpreted as good owner awareness and education. However, this figure may not be a true representation of compliance with ocular medication and would need to be compared to a non-surgical MC group in a prospective study.

From the owner’s perspective, the majority of BOS clinical signs were perceived to have reduced following the MC surgery, but epiphora was not completely resolved, which differs from the outcomes reported by Yi et al. (2006) [15]. The improvement but persistence of epiphora in the current study may be explained by the anatomical repositioning of the lacrimal puncta associated with MC surgery; however, overall tear transition anatomy remains disturbed in extreme brachycephalic dogs and is not corrected with MC [54]. Furthermore, a large proportion of the dogs with epiphora post-MC continue to receive cyclosporine A, which has a lacrimomimetic effect [55].

Corneal ulcerations are considered most detrimental to patient quality of life given the pain associated with it [4]. Consequently, a surgical procedure that significantly reduces the occurrence of corneal ulceration in predisposed populations offers a substantial welfare opportunity and suggests the value of the increased awareness of MC.

Although there was a statistically significant reduction in corneal ulceration risk between pre- and post-operative surgical MC patients based on owners’ reports, long-term follow-up information of ulcer occurrence in the non-surgical MC group would have strengthened the evidence supporting this hypothesis. This information was not available in the present study. A future prospective study with planned re-examinations in both surgical and non-surgical MC patients would increase the statistical power, the lack of which is a limitation of the current study.

Packer et al. (2015) predicted that a reduction in palpebral fissure following MC in dogs with excessively large palpebral fissures should reduce the proportional risk of corneal ulceration, which was supported by the present study [2].

The infantile feature of large, low-lying eyes in brachycephalic dogs has been demonstrated to increase their attractiveness to humans, particularly to women [34,56]. Similar to results reported by Steinmetz et al. (2017), it was reassuring that a large proportion of owners were satisfied with the cosmetic outcome of MC (87.5%, 56/64) despite the change in the appearance of their dog’s eyes [57], with the pre-existing dog–owner bond (which is reported to be high with brachycephalic breeds, particularly with Pug owners [23]) possibly making relatively minor changes in their dog’s appearance more tolerable. This is also potentially combined with psychological trade-offs between an owner’s ‘like’ of their dog’s appearance with their ‘dislike’ of its clinical consequences (if this connection had been made mentally).

Patients referred for specialist veterinary care are not a true representation of the general population, as socio-economic factors will have an impact [58]. In addition, those owners that purchase brachycephalic dogs are reportedly highly influenced by appearance [36], which may consequently affect the owner’s decision making on whether to pursue surgery that will reduce their dog’s extreme appearance towards that of a typical canine [36,47].

The owner response rate of 64% in the current study was comparable to that previously reported by veterinary-based questionnaires [59]. The current survey also excluded dogs that were deceased because of potential distress to owners following recommendation by the Royal Veterinary College Social Science Research Ethical Review Board. This may have biased the overall results. The questionnaire took an average six minutes to complete, which is below the recommended time limit to maximise participation [60]. Requesting recall by owners of fine-detailed information from up to five years ago is likely to have reduced the accuracy of the information provided. Owners’ interpretation and awareness of certain clinical signs may also be variable, particularly chronic abnormal clinical signs [49]. Owner assessment of ocular discharge may also reflect improved education towards periocular cleaning [61].

A further limitation which will have influence on the results are patients who underwent additional ocular procedures, such as removal of distichia or ectopic cilia (7/118, 5.9%). These ocular abnormalities are known to cause ocular irritation, which may have been cumulative with the BOS-associated ocular signs improved by MC [62,63]. Atopic dermatitis and allergic conjunctivitis and their impact on ocular discharge, irritation and corneal ulceration was not a focus of the current study and should be considered in the future. However, 40.6% of owners (26/64) reported persistent ocular irritation post-operatively and, of these, 26.9% (7/26) had been diagnosed with atopic dermatitis; whether this is related to an underlying allergic disease warrants further investigation. Pre-surgical ocular discharge and conjunctival hyperaemia in non-ulcerated dogs with atopic dermatitis were similar to dogs without atopic dermatitis (Appendix A).

A prospective study investigating periocular and conjunctival cytology pre- and post-MC, alongside histopathology and an owner questionnaire, would improve the understanding of MC in dogs with and without atopy. Other skin disease such as persistent nasal fold dermatitis, not addressed by MC, and secondary dermatitis due to persistent epiphora will likely contribute to facial pruritis post-surgery and should be considered in future studies.

In addition, patients who continue to receive long-term immunomodulatory topical medication due to qualitative or quantitative tear film deficiencies or allergic eyelid/conjunctival disease will also show alleviation of ocular discomfort [64]. Both factors will have an impact on ocular health long-term, thus limiting our ability to identify changes solely associated with MC. The number of non-surgical MC dogs that were managed for further corneal ulceration by their primary-care veterinarian is unknown. Long-term follow-up of the non-MC dogs that did not undergo surgery yet continued to receive long-term immunomodulatory topical medication would strengthen this study.

## 5. Conclusions

In summary, this study demonstrates a reduction in the prevalence of corneal ulceration, ocular discharge and ocular irritation in brachycephalic dogs following MC undertaken to reduce the clinical impacts of BOS in dogs. However, multi-modal management for dogs with BOS should be considered. Although chronic ocular abnormalities may remain following surgery, an improvement in associated ocular clinical signs is generally apparent, whereas the impact of concurrent dermatitis warrants further investigation. The majority of clients were satisfied with the clinical and cosmetic outcome following MC and would recommend the surgery despite the requirement for long-term topical medication. These results support the proposal that MC offers a valuable surgical option for BOS in dogs and can substantially redress the negative health and quality-of-life impacts from BOS in breeds with extreme brachycephaly. This study highlights the value of raising awareness in owners, breeders and primary-care veterinarians about BOS and MC.

## Figures and Tables

**Figure 1 animals-13-02032-f001:**
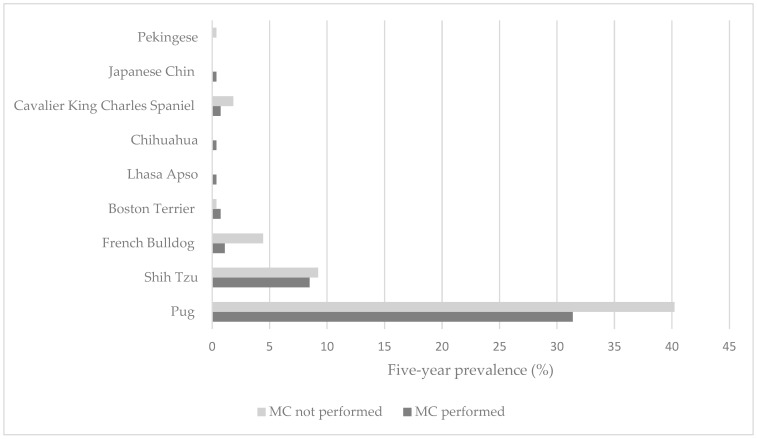
Proportional update of surgery in brachycephalic breeds recommended medial canthoplasty (MC) over 5 years (2016–2021) under referral veterinary care at the Queen Mother Hospital for Animals, Royal Veterinary College, UK.

**Figure 2 animals-13-02032-f002:**
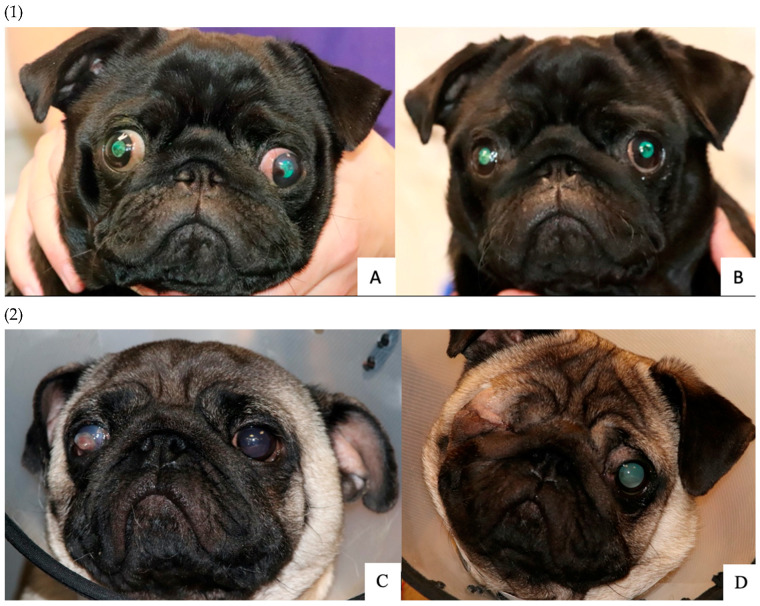
(**1**) Photographs of a Pug presented to the Queen Mother Hospital for Animals, Royal Veterinary College, UK. The patient underwent bilateral medial canthoplasty at one year and three months of age. Image (**A**) shows euryblepharon and increased scleral show, lower medial entropion and corneal pigmentation. A historical corneal ulcer can be seen in the axial cornea of the left eye. Image (**B**) demonstrates improvement in scleral show and corneal pigmentation and resolution of entropion five months after surgery. (**2**) Photographs of a Pug presented to the Queen Mother Hospital for Animals, Royal Veterinary College, UK. The patient underwent unilateral medial canthoplasty of the left eye and enucleation of the right eye at three years and seven months of age. Image (**C**) shows corneal perforation of the right eye and bilateral euryblepharon, mildly increased scleral show and lower medial entropion. Image (**D**) demonstrates improvement in scleral show and eyelid position the day after surgery. ©Royal Veterinary College.

**Figure 3 animals-13-02032-f003:**
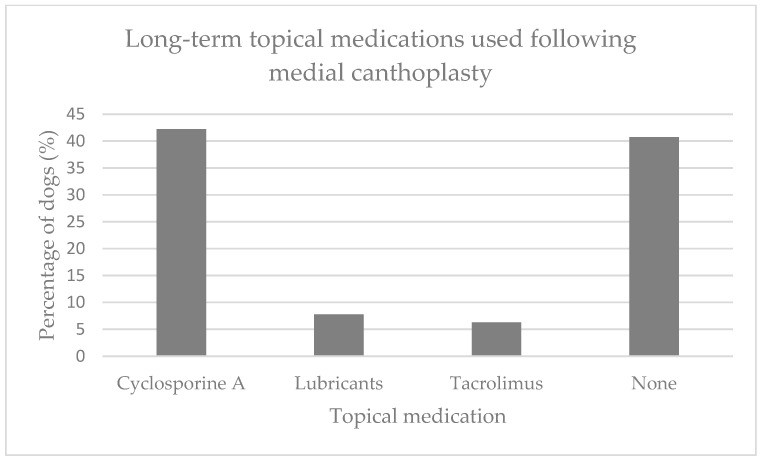
Percentage of patients (*n* = 64) that have remained on long-term topical ocular medication following medial canthoplasty at the Queen Mother Hospital for Animals, Royal Veterinary College, UK, between 2016 and 2021. These results were based on owner questionnaire.

**Table 1 animals-13-02032-t001:** Breeds of dogs with brachycephaly investigated for medial canthoplasty recommendation from 2016–2021 under referral veterinary care at the Queen Mother Hospital for Animals, Royal Veterinary College, UK.

Breeds
Boston Terrier
Cavalier King Charles Spaniel
Chihuahua
English Bulldog
French Bulldog
Japanese Chin
Lhasa Apso
Pekingese
Pug
Shih Tzu

**Table 2 animals-13-02032-t002:** Number and percentage (%) of brachycephalic dog breeds recommended medial canthoplasty out of the total number of brachycephalic breeds presented to the ophthalmology service at the Queen Mother Hospital for Animals, Royal Veterinary College, UK (2016–2021).

Breed	Number Recommended MC */Ophthalmology Referral Population	% of Total Number of Breeds Recommended MC *
Pug	194/275	70.5%
Japanese Chin	1/3	33.3%
Shih Tzu	48/217	22.1%
Pekingese	1/10	10.0%
Cavalier King Charles Spaniel	7/81	8.6%
Boston Terrier	3/36	8.3%
French Bulldog	15/210	7.1%
Lhasa Apso	1/34	2.9%
Chihuahua	1/65	1.5%

* MC Medial canthoplasty.

**Table 3 animals-13-02032-t003:** Descriptive and univariable logistic regression results for age, sex, neuter status and breed as a risk factor for recommendation of medial canthoplasty between 2016 and 2021 in dogs with brachycephaly under referral veterinary care at the Queen Mother Hospital for Animals, Royal Veterinary College, UK. Breeds with low sample size (*n* = 1) were excluded from the model.

Variable	Category	Number of Dogs (%)	OR *	95% CI **
Age (years)	<1.0	38/271 (14.0%)	Base	
	1.0–<2.0	43/271 (15.9%)	0.792	0.330–1.900
	2.0–<4.0	64/271 (23.6%)	0.829	0.371–1.852
	4.0–<6.0	48/271 (17.8%)	0.846	0.361–1.985
	6.0–<8.0	38/271 (14.0%)	0.583	0.233–1.458
	8.0–<10.0	29/271 (10.8%)	0.611	0.229–1.634
	>10.0	11/271 (4.1%)	0.571	0.143–2.279
Sex	Female	115/271 (42.4%)	Base	
	Male	156/271 (57.6%)	1.330	0.817–2.174
Neuter status	Entire	113/271 (41.7%)	Base	
	Neutered	158/271 (58.3%)	1.120	0.685–1.819
Breed	Boston Terrier	3/271 (1.1%)	Base	
	Cavalier King Charles Spaniel	7/271 (2.6%)	0.200	0.011–3.661
	French Bulldog	15/271 (5.5%)	0.125	0.008–1.885
	Pug	194/271 (71.6%)	0.390	0.035–4.372
	Shih Tzu	48/271 (17.7%)	0.460	0.039–5.418

OR * odds ratio, CI ** confidence intervals.

**Table 4 animals-13-02032-t004:** Descriptive results (*n* = number of dogs, % = percentage affected within the specific group) for ocular abnormalities reported in Pugs and non-Pug dogs (Shi Tzu, French Bulldog, Cavalier King Charles Spaniel, Boston Terrier, Chihuahua, Lhasa Apso, Japanese Chin and Pekingese) with recommended medial canthoplasty between 2016 and 2021 under referral veterinary care at the Queen Mother Hospital for Animals, Royal Veterinary College, UK. These ocular abnormalities were categorised in dogs with medial canthoplasty (Surgical MC group *) performed at a later stage or did not pursue MC (Non-Surgical MC group **).

	Surgical MC Group *				Non-Surgical MC Group **			
	Pug		Non-Pug		Pug		Non-Pug	
	n/85	%	n/33	%	n/109	%	n/44	%
Corneal pigmentation	55	65	9	27	90	83	13	30
Entropion	57	67	18	55	102	94	28	64
Euryblepharon with increased scleral show	43	51	17	52	79	72	32	73
Distichia	5	6	5	15	12	11	6	14
Quantitative or qualitative tear film deficiency	8	9	4	12	23	21	7	16
Caruncular trichiasis	0	0	9	27	0	0	14	32
Nasal fold trichiasis	3	4	5	15	2	2	3	7
Eyelid trichiasis	12	14	8	24	18	17	11	25
Corneal ulceration	22	26	7	21	52	48	16	36
Epiphora	2	2	5	15	0	0	3	7
Mucoid ocular discharge	34	40	6	18	0	0	16	36
Blepharospasm/increased blink rate	12	14	1	3	0	0	10	23
Conjunctival hyperaemia	18	21	7	21	0	0	21	48
Ectopic cilia	0	0	0	0	0	0	2	5
Lagophthalmos	3	4	2	6	0	0	6	14
Prominent nasal fold	9	11	0	0	5	5	0	0

**Table 5 animals-13-02032-t005:** Number and percentage of dogs presented to the Queen Mother Hospital for Animals, Royal Veterinary College, UK, between 2016 and 2021 with clinical signs suggestive of ocular abnormalities before and after medial canthoplasty, reported by their owner. * indicates statistical significance between the occurrence pre- and post-operative percentage.

	Number of Dogs Pre-Operative (%)		Number of Dogs Post-Operative (%)	Number of Dogs with Improvement of a Clinical Sign (%)	Number of Dogs with Resolved Clinical Sign (%)	Combined Number of Dogs with Improved and Resolved Clinical Sign (%)	*p*-Value
Presence of ocular discharge	48/64 (75.0%)	Daily 30/48 (62.5%)	42/64 (65.6%)	24/48 (50.0%)	10/48 (20.8%)	34/48 (70.8%)	* *p* < 1 × 10^−5^
		Several times a week 8/48 (16.7%)					
		Weekly 3/48 (6.3%)					
		Monthly 2/48 (4.2%)					
		Rarely 4/48 (8.3%)					
Requirement for periocular cleaning	61/64 (79.7%)	Daily 26/61 (42.6%)	47/64 (73.4%)	22/61 (36.1%)	10/61 (16.4%)	32/61 (52.5%)	* *p* < 1 × 10^−5^
		Several times a week 12/61 (19.7%)					
		Weekly 11/61 (18.0%)					
		Monthly 1/61 (1.6%)					
Ocular irritation	31/64 (48.4%)	Daily 22/31 (71.0%)	26/64 (40.6%)	15/31 (48.4%)	6/31 (19.4%)	21/31 (67.7%)	* 0.030
		Several times a week 8/31 (25.8%)					
		Weekly 1/31 (3.2%)					
		Monthly 1/31 (3.2%)					
History of corneal ulceration	44/64 (68.8%)	One episode 25/44 (56.8%)	8/64 (12.5%)	2/44 (4.5%)	36/44 (81.9%)	38/44 (86.4%)	* *p* < 1 × 10^−5^
		Two episodes 8/44 (18.2%)					
		Three episodes 5/44 (11.4%)					
		Four episodes 2/44 (4.5%)					
		Five episodes 1/44 (2.3%)					

**Table 6 animals-13-02032-t006:** Owner awareness of ocular abnormalities in dogs with brachycephaly and their own dog prior to presentation (pre-operatively). The participants were owners of dogs with brachycephaly which underwent medial canthoplasty at the Queen Mother Hospital for Animals, Royal Veterinary College, UK, between 2016 and 2021.

	Yes	No
Owner aware of ocular disease associated with brachycephaly	33/64 (51.6%)	31/64 (48.4%)
Owner aware of their dog’s ocular abnormalities	13/64 (20.3%)	51/64 (79.7%)

**Table 7 animals-13-02032-t007:** Number and percentage of owner satisfaction following medial canthoplasty surgery in their dog (post-operatively) at the Queen Mother Hospital for Animals, Royal Veterinary College, UK, between 2016 and 2021, and their consideration for medial canthoplasty for their own dog or another dog with brachycephaly in the future.

Question	Number and Percentage of Owner Satisfaction (%)
Owner satisfied with clinical outcome	55/64 (85.9%)
Owner satisfied with cosmetic outcome	56/64 (87.5%)
Owner would consider for another dog with brachycephaly in the future	55/64 (85.9%)
Owner would recommend for another dog with brachycephaly in the future	50/64 (78.1%)

## Data Availability

Data is unavailable in accordance with consent provided by participants and due to ethical restrictions.

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
