# Peer review of "A Review of Clinical Outcomes, Owner Understanding and Satisfaction following Medial Canthoplasty in Brachycephalic Dogs in a UK Referral Setting (2016–2021)"

_animals, 2023, doi:10.3390/ani13122032_

Round 1

Reviewer 1 Report

This is an interesting manuscript, which looked in the surgical causes, outcomes, and client satisfaction of medical canthoplasty procedures in brachiocephalic dogs.

Minor issues: 

Introduction - paragraph 2: While corneal ulceration and pigmentary keratitis are common reasons for medial canthoplasty (MC), in this reviewer's opinion, exposure keratitis, risk of the globe proptosis or previous incidence of the subclinical and clinical globe proptosis, and facail/palpebral nerve paralysis are equally important indications for pursuing MC in brachiocephalics.

Major issue:

The major issue which significantly impacts the quality of the manuscript is complete ignorance of the issue of allergic skin disease. Based on the literature data and reviewer's personal experience Pug as a breed is among top 5 breeds affected with atopic dermatitis ( Mazrier et al, Vet Dermatol2016 Jun;27(3):167-e42. doi: 10.1111/vde.12317.). Our standard clinical practice for all surgical entropion and medical canthoplasty procedures is a histopathological evaluation of the resected skin/conjunctiva, and almost 95% of all of our histo specimens have evidence of the allergic eyelid/conjunctiva disease.

Even the author's own data in the results section (3.2.2) reports 40% incidence of periocular irritation in all operated patients, further validating the hypothesis that pruritus remains a significant clinical component regardless of the successful surgery.

Unfortunately, the questionnaire studies have its own limitation, but questions pertinent to the clinical signs of allergies were completely committed. 

I would suggest that authors carefully review clinical records of their patients, and report whether any of patients were diagnosed with any form of the ocular/skin allergic disease or receiving the treatment.

This reviewer would recommend introducing a data set on conjunctival hyperemia/mucoid discharge in non-ulcerated eyes with percentages of incidence, which will likely point to the importance of allergic periocular disease in this patient population

The discussion as is seems to be too long, and poorly focused, so the manuscript may benefit significantly from shortening of the discussion.   

Author Response

Dear Reviewer,

Many thanks for your comments and guidance. Please find the responses and references below (continuous page numbering system used):

Minor issues: 

Introduction - paragraph 2: While corneal ulceration and pigmentary keratitis are common reasons for medial canthoplasty (MC), in this reviewer's opinion, exposure keratitis, risk of the globe proptosis or previous incidence of the subclinical and clinical globe proptosis, and facial/palpebral nerve paralysis are equally important indications for pursuing MC in brachiocephalics.

Thank you for your comment. It is a very interesting aspect. We reviewed the literature again and were not successful finding a reference for BOS and MC in the context of facial nerve paralysis and would therefore not include this aspect in the introduction. The aspect of proptosis is included in paragraph 3 (page 2,line 295) and subclinical proptosis was added (page2, line 295)

Major issue:

The major issue which significantly impacts the quality of the manuscript is complete ignorance of the issue of allergic skin disease. Based on the literature data and reviewer's personal experience Pug as a breed is among top 5 breeds affected with atopic dermatitis (Mazrier et al, Vet Dermatol. 2016 Jun;27(3):167-e42. doi: 10.1111/vde.12317.). Our standard clinical practice for all surgical entropion and medical canthoplasty procedures is a histopathological evaluation of the resected skin/conjunctiva, and almost 95% of all of our histo specimens have evidence of the allergic eyelid/conjunctiva disease.

Thank you for your highly valuable comment. The retrospective data were reviewed and it has been ensured to implement the factor of atopic dermatitis throughout the revised manuscript. The introduction was changed accordingly and a reference included (page 2, line 329-331). Thank you for sharing your clinical expertise and standard of clinical practice. We would have liked to include a reference of published material regarding allergic skin/ocular disease and eyelid correction, however, this aspect was included in the discussion (page 17, line 2667-2672).

Even the author's own data in the results section (3.2.2) reports 40% incidence of periocular irritation in all operated patients, further validating the hypothesis that pruritus remains a significant clinical component regardless of the successful surgery.

Thank you for your comment. This has now been addressed more thoroughly throughout the manuscript (page 12, line 1274-1281; page 17, line 2637-2642).

Unfortunately, the questionnaire studies have its own limitation, but questions pertinent to the clinical signs of allergies were completely committed. 

The limitation and future study design is discussed (page 17, line 2692-2694). We are unable to contact the clients again due to the ethical board agreement so unfortunately despite the validity of this point, it cannot be explored further with the owners that completed the questionnaire.

I would suggest that authors carefully review clinical records of their patients, and report whether any of patients were diagnosed with any form of the ocular/skin allergic disease or receiving the treatment.

Thank you for highlighting this very important point. The clinical history of all dogs has been reviewed and this information is now included (page 4 line 432, page 7 line 646-649, page 12, line 1274-1276 and 1280, page 17 line 2687-2689).

This reviewer would recommend introducing a data set on conjunctival hyperemia/mucoid discharge in non-ulcerated eyes with percentages of incidence, which will likely point to the importance of allergic periocular disease in this patient population.

Thank you for this comment. The retrospective data were reviewed. Please find the results below. There was no obvious trend in increased mucoid discharge and conjunctival hyperaemia in the absence of corneal ulceration in patients with atopic dermatitis. However, with a higher power these difference might become significant (in future studies). The discussion was changed accordingly (see page 17, line 2683-2685).

Total study population (pre recommended MC*) 

Atopic dermatitis 

Non atopic dermatitis 

Ulcerated 

Non-ulcerated 

Ulcerated 

Non-ulcerated  

Ulcerated  

Non-ulcerated  

n/X 

% 

n/X 

% 

n/X 

% 

n/X 

% 

n/X 

% 

n/X 

% 

Mucoid ocular discharge 

12/25 

48.0 

 29/246

11.8

 1/4

25.0 

 12/38

31.6 

2/21 

9.5 

60/208 

28.8 

Conjunctival hyperaemia 

 8/25

32.0 

 21/246

8.5 

 1/4

25.0 

15/38 

39.5 

3/21 

 14.3

 46/208

22.1 

The importance to consider allergic aetiology for patients with facial pruritus is important however, mucoid discharge and conjunctival hyperaemia is a rather unspecific clinical sign and needs to rule out lot of other ophthalmic disease. Conjunctival chemosis is a more specific clinical sign, at least for allergic conjunctivitis (not atopic dermatitis) but was not observed in any patients in this sutdy. (BMC Vet Res. 2023 Feb 3;19(1):35.doi: 10.1186/s12917-022-03561-5. Diagnostic approach and grading scheme for canine allergic conjunctivitis)

The list of differential diagnosis for mucoid discharge and conjunctival hyperaemia is extensive and some, but not all of them are considered in the current study.
(Mechanical irritation (ectopic cilia, distichiasis), precorneal tearfilm abnormalities (i.e. quantitative&qualitative keratoconjunctivits sicca), conformational such as nasal pocket syndrome (dolichocephalic breeds), absence of nasal tear transition (as in brachycephalic dogs) with increased secondary bacterial conjunctivitis, lagophthalmos with chronic exposure keratitis and conjunctivitis and chronic pain, entropion with hair touching the cornea leading to chronic/pigmentary keratitis as in most BOS patients, follicular conjunctivitis in young dogs).

As the aim of the present study was to identify the surgical outcomes, complications, and incidence of corneal ulceration before and after the MC and to explore owner awareness and satisfaction following this surgery, we still hope to have addressed the raised point and the reviewer consider it as acceptable that the additional data is placed in the supplementary (table 1).

The discussion as is seems to be too long, and poorly focused, so the manuscript may benefit significantly from shortening of the discussion.   

Thank you for this valid point. The discussion has been significantly reduced in length.

Kind regards,

The Authors

Reviewer 2 Report

Dear Authors, the manuscript is very clear and exhaustive, you have done a great deal of data collection and the analysis of these provides a complete picture of the effects of medial canthoplasty surgery in canine brachycephalic ocular syndrome. The paper deserves to be published in Animals, I have only a few minor revisions: 

1. The title needs to be changed.

From the title it seems to be dealing with an experimental research, while it is a retrospective survey on patients who were recommended Medial canthoplastic surgery from 2016 to 2021. This must be clear to the reader right from the title.

2. Remove the term "welfare" from the title as well as throughout the text.

The use of "welfare" seems inappropriate as it expresses a much more complex concept than a simple improvement in the dog's quality of life. Indeed, no complete information concerning animal welfare could be extracted from the used data.

3. The simple summary must be rewritten to make it "simpler". The simple summary should be written for a lay reader, thus, it should contain only a clear statement of the problem being addressed, the goals and objectives, relevant findings, conclusions of the study and how they will be valuable to society. Please check the instruction for authors for details and examples.

Once this is done, everything is fine for me.

Author Response

Dear Reviewer,

Many thanks for your comments and guidance. Please find the responses below (continuous page numbering system used):

  1. The title needs to be changed.

From the title it seems to be dealing with an experimental research, while it is a retrospective survey on patients who were recommended Medial canthoplastic surgery from 2016 to 2021. This must be clear to the reader right from the title.

Thank you. The title has been changed to reflect the retrospective nature of the study.

  1. Remove the term "welfare" from the title as well as throughout the text.

The use of "welfare" seems inappropriate as it expresses a much more complex concept than a simple improvement in the dog's quality of life. Indeed, no complete information concerning animal welfare could be extracted from the used data.

Thank you for this comment. The word welfare was changed to 'quality of life' in the majority of the manuscript (page 1 line 24/41; page 2 line 292/304; page 3 line 355/357; page 16, line 2199; page 17 line 2710), unless 'welfare' was referring to specific literature.

The simple summary must be rewritten to make it "simpler". The simple summary should be written for a lay reader, thus, it should contain only a clear statement of the problem being addressed, the goals and objectives, relevant findings, conclusions of the study and how they will be valuable to society. Please check the instruction for authors for details and examples.

Thank you for your comment. Some of the more detailed information has been removed and the remainder was reworded.

Once this is done, everything is fine for me.

Kind regards,

The Authors

Reviewer 3 Report

I attach a Word document

Author Response

Dear Reviewer,

Many thanks for your comments and guidance. Please find my response and references below (continuous line number system used):

M1: The results from line 36 to 39 are confusing. Please, explain to which group the 64 and 104 patients belong to. The number and percentages of animals for the different clinical signs are confusing. Where do these results came from?

Thank you for this comment. To improve clarity, it is now stated that 104 dogs returned for follow-up and 64 owners completed the questionnaire (page 1, line 35/36).

M2: It must go in last position (alphabetical order).

This has been changed (page 1, line 43).

M3: Please write it more clear. Not use ";" . I suggest to write "BOAS: brachycephalic obstructive airway syndrome"

This has been altered as per your request (page 1, line 44/45).

M4: Please make attention at this statement. In the dogs euthanized in that study the ocular disease was not the main problem to request the euthanasia, there were other clinical problems that compromised the welfare of the animals.

This has been changed (page 2 line 303).

M5: Specify the type of surgeries for corneal ulcers.

This has been changed with added information (page 2 line 287-288).

M6: You have nominated it before, so it's not necessary to explain it again

Thank you for bringing this to our attention, it has been deleted (page 2 line 281).

M7: As this is the first time that you have nominated medial canthoplasty, you have to write "medial canthoplasty (MC)"

This has been changed (page 2, line 297).

M8: Delete (MC)

This has been deleted (page 2, line 295).

M9: There are many repetition of "medial canthoplasty"

This has been changed throughout the manuscript.

M10: This is an article, not a book. It's not necessary to write all the definitions of entropion, euryblepharon, trichiasis, lagophthalmos, epiphora and corneal ulceration.

Thank you for your comment, this has been deleted (page 4).

M11: The graphic in the Figure 1 has not been design well. For any breed you have to create two bars (MC performed and MC not performed), one underneath the other for explain the proportional update.

Thank you. This has been changed as requested (page 6, line 582).

M12: Decide how to write the number of cases in the manuscript: number and then the percentage or the opposit, and then apply in all the document. There must be a rule to be apply in the entire document.

This was revised and remains consistent throughout the revised manuscript. Thank you.

M13: Did the dogs underwent MC in another place? How long after your recommendation? Because if the dogs have done the surgery in your center, did you have considered them within the MC group?

Thank you for bringing this to our attention. Patients underwent MC represent our own hospital population. The sentence has been changed to provide clarification (page 8, line 686-688)

M14: Specify the concentration of cyclosporine and tacrolimus used.

These have been added throughout the revised manuscript (page 10 line 1031-1032/1036-1037, page 13 line 1509-1510).

Kind regards,

The Authors

Round 2

Reviewer 1 Report

The authors adequately addressed raised criticisms, and this reviewer does not have any additional objections.